# Skilled birth attendance and its associated factors in Chad and Nigeria: A multilevel analysis of DHS data

Wubshet D. Negash[1]*, Henok Dessie Wubneh[2]

1 Department of Health Systems and Policy, Institute of Public Health, College of Medicine and Health Sciences, University of Gondar, Gondar, Ethiopia, 2 National Centre for Epidemiology and Population Health, The Australian National University, Canberra, Australia

* wubshetdn@gmail.com

## Abstract

In the year 2020, Worldwide, 211 maternal deaths occurred per 100,000 livebirths. In particular, Chad and Nigeria report extremely high (>1000) maternal mortality. Despite this, there has been limited research on skilled birth attendance (SBA) and its factors in Chad and Nigeria. Therefore, this study aimed to assess the prevalence of SBA and associated factors among women of reproductive age in extremely high maternal mortality countries. This study is a secondary data analysis based on the Demographic and Health Survey Chad 2014–15, Nigeria 2018, involving 52,666 study participants. We used Stata version 17.0 to analyze the data. A mixed-effects binary logistic regression model was employed to account for the hierarchical structure of the data. An odds ratio along with a 95% CI were generated to identify factors associated with SBA. A p-value less than 0.05 was declared as statistically significant. In this study, the prevalence of SBA among reproductive-age women was 41.90% (95% CI: 41.48 - 42.32). ANC visits (AOR = 5.56; 95% CI: 5.03 - 6.14), Primary (AOR = 1.77; 95% CI: 1.62 - 1.95) and secondary education (AOR = 4.06; 95% CI: 3.59 - 4.57), middle (AOR = 1.37; 95% CI: 1.23 - 1.52) and rich (AOR = 2.77; 95% CI: 1.87-2.38) wealth categories, media exposure (AOR = 1.50; 95% CI: 1.38 -1.63), and community-level education (AOR = 2.73; 95% CI: 2.26 - 3.29) were significant factors associated with skilled birth attendance. 41.90% of reproductive-age women had SBA in countries with high burden maternal mortality. Education, wealth index, media exposure, ANC visit, distance to the health facility, and place of residence were factors for the assisted birth attendance. Therefore, the respective country governments should work on women's and community education, extensive ANC visits, and media exposure.

**Data availability statement:** The data used in this study are publicly available from the demographic and health surveys (DHS) program and attached as supplementary file 1. Access requires a formal request through the DHS website: https://dhsprogram.com/data.

**Funding:** The author(s) received no specific funding for this work.

**Competing interests:** The authors have declared that no competing interests exist.

## Introduction

There is widespread recognition of the importance of skilled birth attendance (SBA), and it is considered imperative to decrease maternal and neonatal mortality [3,4]. The health of women during pregnancy and childbirth is particularly important, especially among adolescents because they exhibit more severe conditions such as maternal anemia, gestational hypertension, malnutrition, eclampsia, preterm birth, and low birth weight [5,6].

In 2017, Worldwide, nearly 300,000 maternal deaths occurred due to complications of pregnancy and delivery. Of the 300,000 deaths, 94% were from low and middle income countries [7]. This prevalence was improved to 260,000 in 2023 with 197 maternal death per 100,000 live births [8]. The most common causes of mortalities among African mothers are hemorrhage (34%), infection (17%), hypertension (9%), obstructed labor (4%), abortion (4%) and anemia (4%) [9]. However, in order to achieve the Sustainable Development Goals (SDGs) target of a global maternal mortality rate (MMR) below 70, there needs to be a reduction of global MMR by 7.5% a year [10].

Many existing studies use pooled SSA data, which may mask country specific, context sensitive factors in countries with high maternal mortality burden [11,12]. Our study focuses on Chad and Nigeria, because these two countries are consistently rank among the highest globally and within SSA in maternal mortality, with rates of 1063 and 1047 per 100,000 livebirths in 2017–2020, respectively [1,2]. In comparison, the maternal mortality rates are significantly low in other SSA countries, For example, 267 in Ethiopia [13], 149 in Kenya [14] and 943 in Ghana [14].

To progress maternal and child health, counter measure policies were implemented at the global and national levels before the Millennium Development Goals (MDGs) era, such as the Safe Motherhood Initiative, the Basic Package of Health Services, and the Essential Package of Health Services (EPHS) [15]. It is estimated that almost 60% of African women give birth without a skilled attendant, leading to high maternal mortality rates. Two out of three women who require emergency obstetric care do not have access to skilled birth attendants [16]. The Sustainable Development Goals (SDGs), adopted at the 2015 United Nations General Assembly, further emphasized reducing maternal mortality. SDG 3.1 specifically calls for a reduction of MMR to less than 70 per 100,000 live births by 2030 [17]. Considering that the majority of maternal deaths and obstetric complications occur during delivery and cannot be predicted beforehand, skilled attendance at birth remains the most effective intervention in reducing maternal mortality [18].

There is a slight increment in SBA from 25.6% in 2011 to 33.1% in 2021 among young women in northern Nigeria [19]. Sub-Saharan Africa (SSA) and South Asia account for 87 percent of maternal deaths and 65 percent of neonatal deaths, respectively [20]. It is possible to prevent one-third of women's deaths and half of all newborn deaths by increasing the number of women receiving skilled delivery assistance [21–23]. Although previous studies tried to assess care by qualified birth attendant in whole sub-Saharan African countries [24,25], there may be overestimation of skilled birth attendance in extremely high mortality countries, focusing on Chad and Nigeria.

Yet, there has been limited research on factors of SBA in countries with high burden maternal mortality. Considering both individual and community-level variables of SBA enables a multilevel approach that can capture broader structural and health system influences often missed in individual-level alone, thereby contributing to the reduction of maternal mortality in Chad and Nigeria and the achievement of SDG 3 [26]. The selection of study variables was informed by their availability in the demographic survey data and prior empirical studies that consistently identify factors such as maternal education, household wealth index, antenatal care visits, parity, and media exposure as critical determinants of maternal health service utilization [27]. Thus, to provide a comprehensive picture of SBA among women of reproductive age in sub-Saharan African countries with extremely high maternal mortality, empirical evaluation is needed to inform up-to-date intervention by respective country governments and other partners working on maternal health. Therefore, this study was aimed to assess the prevalence of assisted delivery by skilled personnel among women of reproductive age in countries with high burden maternal mortality, focusing on Chad and Nigeria to provide a more nuanced understanding of both individual- and community-level factors. This paper provides evidence-based interventions to address significant factors related to delivery with skilled birth attendant to inform governments, policymakers, and other stakeholders in the corresponding countries.

## Methods

### Study setting and designs

We conducted the study in extremely high maternal mortality countries with a specific focus on Chad and Nigeria in the year 2014/15 and 2018, respectively. We used the demographic health survey data. The United Nations International Children's Fund (UNICEF) reported that South Sudan, Chad, and Nigeria had extremely high maternal mortality rates (>1000) in 2020 [1,2]. According to WHO estimates for 2017–2020, Chad had an MMR of 1,063 per 100,000 live births and Nigeria 1,047 per 100,000 live births [1,2], consistent with the period corresponding to the most recent DHS survey data referenced in this study. More recent 2023 estimates indicate a slight decline, with Nigeria at 993 and Chad at 748 per 100,000 live births. We purposely excluded South Sudan because of no DHS data [28]. Data from the DHS program's official database can be accessed at https://dhsprogram.com by requesting authorization online.

Community-level clusters were defined using the DHS primary sampling units (PSUs), which correspond to census enumeration areas. Each cluster has a unique identifier (V001). These clusters, stratified by region and urban/rural residence, were used as the community-level unit in the multilevel analysis to account for contextual variation in skilled birth attendance. The Kids Record (KR) data set was used to extract all the factors associated with skilled birth attendance. Additionally, the KR data set provides comprehensive information on child health outcomes, immunization status, maternal care practices, allowing for more detailed assessment of maternal and child health service utilization and program planning. The nationally representative data of the demographic health survey, which has been conducted every five year interval across African countries [29]. A two-stage stratified sampling technique was used [29]. For the analysis, a total weighted sample of 52,666 reproductive-age women was used (S1 Data).

### Study variables

In line with the WHO definition, skilled birth attendants in this study are defined as health professionals including doctors, nurses, midwives, health officers, and health extension workers who are trained to manage normal pregnancies, childbirth, and the immediate postnatal period, as well as to recognize and manage complications. Skilled birth attendance was the outcome variable, coded as "1" if the birth was attended by any of these professionals and "0" otherwise [30]. Other independent variables are age (15–24, 25–34, 35 & above), residence (urban, rural), educational status (no formal education, primary education, secondary and higher education), wealth index (poor, middle, and rich), current marital status (married, unmarried), and media exposure (those women who were either reading newspapers/magazine, or listening to radio and watching television at least once a week were considered as having media exposure), those with no access to

any were considered unexposed. The community-level education, media exposure and poverty were designed by combining their corresponding individual level variables at the cluster level. Finally, the variables were categorized by using the median into high and low, since these were not normally distributed. The median approach allows for consistent categorization across settings, supports comparability, and simplifies interpretation of community effects. The community-level education was generated by the proportion of households in the educated categories. Categorized as low if the proportion of women were educated was below 50% and high if the proportion is ≥ 50% [31,32]. Community-level poverty was aggregated by the proportion of households in the poorest and poorer quintile. Grouped as low if the proportion from a given community is < 50% and high if the proportion is ≥ 50%. Community level media exposure was generated by the proportion of media exposure. Categorized as low if the proportion from a given community is < 50% and high if the proportion is ≥ 50% [31,32].

## Modeling approaches

The demographic health survey data follows a hierarchical structure (individuals nested within communities or clusters), which violates the assumptions of common logistic regression analysis, specifically, the independence of observations and equal variance. In this context, women are grouped within clusters, based on the idea that those within the same cluster tend to share similar characteristics. Therefore, more advanced statistical models are required to appropriately account for variations between clusters. We used a multilevel binary logistic regression model to identify both individual and community-level factors of SBA among reproductive-age women in countries with extremely high burden of maternal mortality. For analysis purposes, we used STATA version 17 software. Weighting (v005/1,000,000) was done to ensure the DHS sample representative and to obtain reliable estimations. All analyses were weighted using the DHS individual sampling weights (v005), which adjust for unequal probabilities of selection and non-response, ensuring nationally representative estimates. The multilevel models incorporated these weights while also accounting for clustering at the community level (PSU) and stratification by region. Prior to model fitting, we assessed multicollinearity among individual- and community-level predictors using Variance Inflation Factor (VIF). All VIF values were below 5, indicating no significant multicollinearity.

Variables that had a p-value less than 0.05 in the bivariable analysis were considered for multivariable analysis. Following the bivariable regression, a multilevel logistic regression analysis comprising both fixed and random effects was conducted. The bivariable model can be expressed as: $\text{logit}(P(Y_i = 1)) = \beta_0 + \beta_1 X_i$. where $\text{logit}(P(Y_i = 1))$: the natural log of the odds that woman i has skilled birth attendance. $Y_i$: outcome variable (1 = skilled birth attendance, 0 = otherwise). $\beta_0$: intercept, representing the log odds of skilled birth attendance when predictors are zero. $\beta_1$: regression coefficient showing the effect of predictor $X_i$ on skilled birth attendance. $X_i$: independent variable for individual i.

The fixed effect of the model was described as an adjusted odds ratio (AOR). The random effects were summarized using the intra-class correlation coefficient (ICC), which indicates the proportion of variance in the outcome attributable to differences between communities, highlighting the influence of community-level factors and justifying the use of multilevel modeling [33]. The null model (Model 0) includes only random intercepts and is used to assess the baseline cluster-level variance in SBA, providing an estimate of how much variation exists between communities before adding explanatory variables. Model I: adjusted for individual-level factors. Model II: adjusted for community-level factors. Model III: adjusted for both individual and community-level factors with the SBA [33,34]. Deviance, or log likelihood was used as a model comparison.

## Ethics statement

To access the data and ethical approval we used Demographic Health Survey website: www.measuredhs.com. In accordance with ethical standards, the consent manuscript was reviewed by the Institutional Review Board of the Demographic and Health Surveys (DHS) program data archivists following submission to the DHS program/ICF International. According

to United States Department of Health and Human Services requirements for the protection of human subjects, all the procedures were approved by ICF International (Ref/209741/24). All subjects and/or their legal guardians of minors under the age of 16 provided informed consent. A third party was not given access to the data set. This study is not experimental. Further explanation of how the DHS uses data and its ethical standards can be found at: http://goo.gl/ny8T6X.

## Results

### Characteristics of individual level factors

Nearly half (49.9%) of the study participants were in the age group of 25–34 years. Of the study participants, 53.1% had no formal education. The majority (70.2%) of the study participants had a history of antenatal care visits (Table 1).

### Community level factors of skilled birth attendance

The majority (68.1%) were rural residents. Of the study participants, 34.1% had a big problem with distance to the health facility. Of the study participants, 51.2% were from communities with a high proportion of media exposure (Table 2).

### Prevalence of skilled birth attendance

The prevalence of skilled birth attendance in extremely high maternal mortality countries is 41.9% (95% CI: 41.48, 42.32) (Fig 1).

### Random effect and model fitness

As observed in Table 3 below, the intra-class correlation coefficient (ICC) for the null model is 61%. This 61% implies that the overall variability of SBA is attributed to cluster variability. Moreover, if we randomly choose two clusters, the odds of skilled birth attendance are 2.69 times higher among women who are from clusters with high skilled birth attendance as compared with their counterparts. The proportional change in variance also increased from 76.77% to 78.54% in the last model. This proportion implies that the last model best displays the institutional delivery variability. Deviance was used

**Table 1. Individual level characteristics of reproductive-age women in extremely high maternal mortality countries (n = 52,666).**

| Variables | Categories | Weighted frequencies | Weighted percentage |
|---|---|---|---|
| **Age of respondents** | 15-24 | 13706 | 26.0 |
| | 25-34 | 26279 | 49.9 |
| | 35 and above | 12681 | 24.1 |
| **Household wealth** | Poor | 23169 | 44.0 |
| | Middle | 10837 | 20.6 |
| | Rich | 18660 | 35.4 |
| **Educational status of the respondent** | No education | 27988 | 53.1 |
| | Primary | 14784 | 28.1 |
| | Secondary & Higher | 9894 | 18.8 |
| **ANC visit** | Yes | 23169 | 70.2 |
| | No | 9839 | 29.8 |
| **Current marital status** | Married | 47327 | 89.9 |
| | Unmarried | 5339 | 10.1 |
| **Media exposure** | Yes | 25913 | 49.5 |
| | No | 26448 | 50.5 |

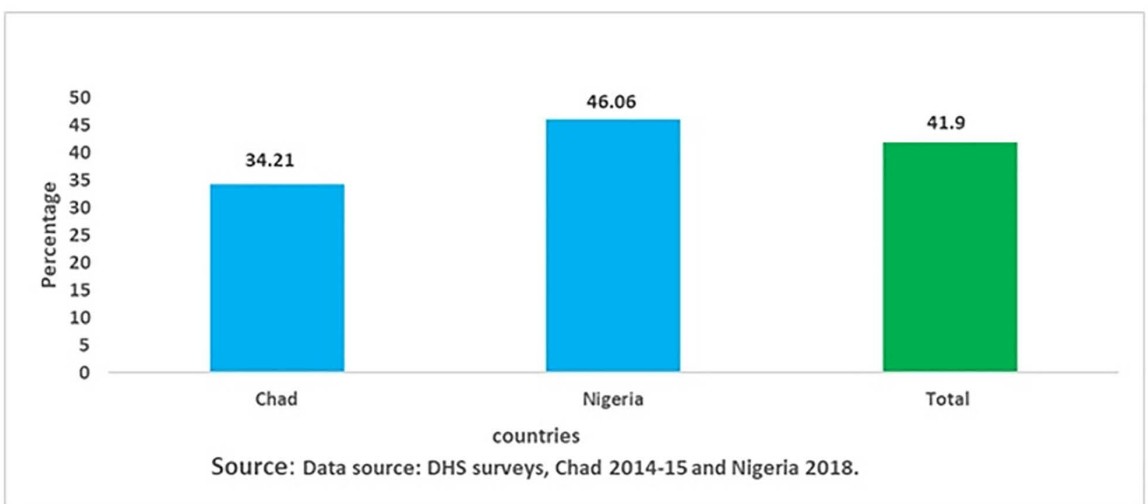

**Table 2. Community level characteristics of reproductive-age women in extremely high maternal mortality countries (n = 52,666).**

| Variables | Categories | Weighted Frequencies | Weighted percentage |
|---|---|---|---|
| **Residence** | Rural | 35881 | 68.1 |
| | Urban | 16785 | 31.9 |
| **Distance to the health facility** | Big problem | 13849 | 34.1 |
| | Not big problem | 26794 | 65.9 |
| **Community level education** | Low | 25529 | 48.5 |
| | High | 27137 | 51.5 |
| **Community level media exposure** | Low | 25702 | 48.8 |
| | High | 26964 | 51.2 |
| **Community level poverty** | Low | 26687 | 50.7 |
| | High | 25979 | 49.3 |

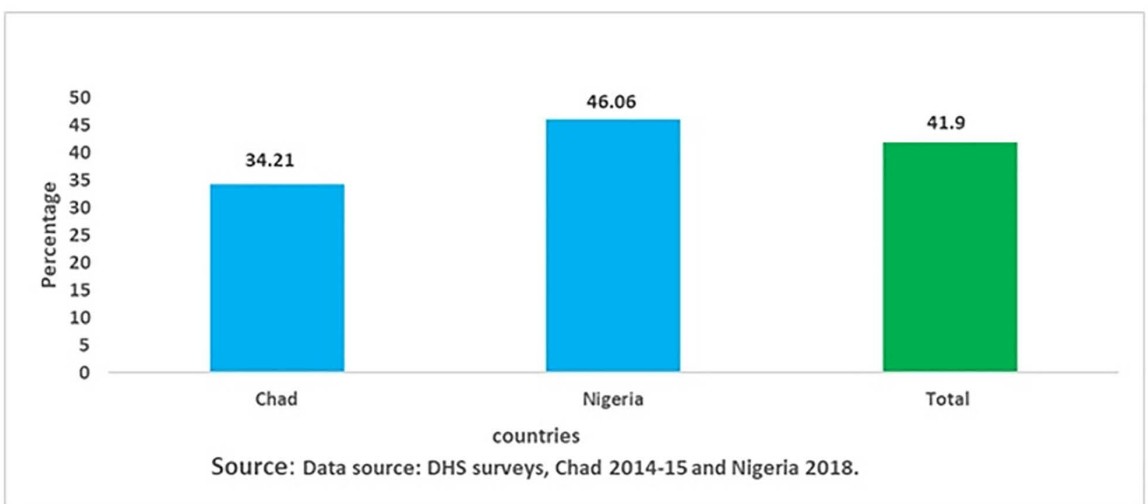

**Fig 1. Prevalence of skilled birth attendance among reproductive-age women in extremely high maternal mortality countries (n = 52,666).**

for model fitness assessment (-2LLR). Model III was found to have the lowest deviance (21971.126), which implies the best-fitting model (Table 3).

### Factors associated with skilled birth attendance among reproductive-age women

The odds of SBA were 1.77 (95% CI: 1.62 - 1.95) and 4.06 (95% CI: 3.59 - 4.57) times higher among women of reproductive age who had primary and secondary education as compared with those study participants with no formal education. Middle 1.37 (95% CI: 1.23 - 1.52) and rich 2.11 (95% CI: 1.87 - 2.38) wealth category women of childbearing age had higher odds of delivery attended by a skilled provider as compared with poor wealth category participants. The odds of SBA were 1.50 (95% CI: 1.38 - 1.63) times higher among media-exposed women in their reproductive years as compared with their counterparts. Those respondents who had history of ANC visit had 5.56 (95% CI: 5.03 - 6.14) times higher odds of SBA as compared with those women of reproductive age who had no history of ANC visit. The odds of SBA were 2.73 (95% CI: 2.26 - 3.29) times higher among study participants who were from high proportion of community level education as compared with their counterparts (**Table 3**).

**Table 3. Multivariable analyses for factors associated with skilled birth attendance (n = 52,666).**

| Categories | Null model | Model 1 AOR (95% CI) | Model 2 AOR (95% CI) | Model 3 AOR (95% CI) |
|---|---|---|---|---|
| **Age** | | | | |
| 15-24 | | 1 | | 1 |
| 25-34 | | 0.99(0.93 - 1.06) | | 0.96(0.88 - 1.05) |
| ≥35 | | 1.16(1.06 - 1.26) | | 1.1(0.99 - 1.20) |
| **Respondent education status** | | | | |
| No education | | 1 | | 1 |
| Primary | | 2.10(1.95 - 2.26) | | **1.77(1.62 - 1.95)** |
| Secondary & Higher | | 5.57(5.01 - 6.20) | | **4.06(3.59 - 4.57)** |
| **Current marital status** | | | | |
| Unmarried | | 1 | | 1 |
| Married | | 0.74(0.67 - 0.81) | | **0.81(0.72 - 1.01)** |
| **Wealth index** | | | | |
| Poor | | 1 | | 1 |
| Middle | | 1.44(1.33 - 1.57) | | **1.37(1.23 - 1.52)** |
| Rich | | 2.65(2.43 - 2.88) | | **2.11(1.87 - 2.38)** |
| **Media exposure** | | | | |
| No | | 1 | | 1 |
| Yes | | 1.66(1.55 - 1.78) | | **1.50(1.38 - 1.63)** |
| **Wanted of pregnancy** | | | | |
| Wanted then | | 1 | | 1 |
| Wanted latter | | 1.04(0.94 - 1.56) | | 0.97(0.86 - 1.01) |
| Wanted no more | | 1.41(1.17 - 1.69) | | **1.31(1.06 - 1.61)** |
| **ANC Visit** | | | | |
| No | | 1 | | 1 |
| Yes | | 5.23(4.82 - 5.69) | | **5.56(5.03 - 6.14)** |
| **Residence** | | | | |
| Urban | | | 1 | 1 |
| Rural | | | 0.21(0.19 - 0.24) | **0.52(0.45 - 0.59)** |
| **Distance to the health facility** | | | | |
| Big problem | | | 1 | 1 |
| Not big problem | | | 1.53(1.43 - 1.64) | **1.13(1.04 - 1.24)** |
| **Community level media exposure** | | | | |
| Low | | | 1 | 1 |
| High | | | 1.36(1.09 - 1.71) | 1.11(0.92 - 1.34) |
| **Community level education** | | | | |
| Low | | | 1 | 1 |
| High | | | 6.58(5.26 - 8.21) | **2.73(2.26 - 3.29)** |
| **Community level poverty** | | | | |
| Low | | | 1 | 1 |
| High | | | 0.45(0.36 - 0.55) | **0.70(0.58 - 0.83)** |
| **Country** | | | | |
| Nigeria | | | 1 | 1 |
| Chad | | | 0.97(0.89 - 1.05) | **1.23(1.09 - 1.38)** |

*(Continued)*

**Table 3.** (Continued)

| Categories | Null model | Model 1 | Model 2 | Model 3 |
|---|---|---|---|---|
| | | AOR (95% CI) | AOR (95% CI) | AOR (95% CI) |
| **Random effect** | | | | |
| Variance | 5.08 | 1.18 | 2.01 | 1.09 |
| ICC | 0.61 | 0.27 | 0.38 | 0.25 |
| MOR | 5.83 | 2.81 | 3.67 | 2.69 |
| PCV(%) | Ref | 76.77 | 60.43 | 78.54 |
| **Model comparission** | | | | |
| Loglikelihood (LLR) | -26831.136 | -14746.844 | -18893.667 | -10985.563 |
| Deviance (-2LLR) | 53662.272 | 29493.688 | 37787.334 | 21971.126 |

ICC = Intra class correlation coefficient MOR = Median odds ratio, PCV = proportional change in variance. AOR = adjusted odds ratio; CI = confidence interval.

## Discussions

Our analysis of the publicly available DHS data from extremely high maternal mortality countries, focusing on Chad and Nigeria depicted that 41.90% (95% CI: 41.48 - 42.32) of childbearing age women had SBA. While the unadjusted prevalence of SBA is higher in Nigeria (Fig 1), the adjusted odds in the multilevel model are higher in Chad. This gap suggests varying individual- and community-level attributes in the two countries. However, even if these factors have been considered, women from Chad still seem to be more likely to use skilled birth attendants, illustrating the significance of multilevel analysis to interpret determinants beyond crude prevalence. It is to note that the DHS for Chad (2014–15) and for Nigeria (2018) were undertaken at different intervals. Accordingly, caution should be taken in interpreting the observed variation in skilled birth attendance across countries, as the time gap may introduce unmeasured confounding. An additional explanation could be due to changes in health systems, policies and coverage of services over time that may have taken place independently from the contextual or individual levels.

The overall magnitude is lower than studies conducted in SSA, where 53% reported skilled birth services [25] and among young women in 29 African countries, where 75% of young women get skilled birth services [35]. This is because of the differences in study setting, where the current study was conducted in extremely high maternal mortality countries in comparison with the whole SSA countries. Moreover, the difference in study participants may have also contributed to the variation in findings. The previous study was conducted among young women as compared with all reproductive-age women in this study. Hence, there is a need to implement programs that discourse the disparities to increase the utilization of skilled birth attendance to reduce complications and maternal mortalities among reproductive-age women in extremely high maternal mortality sub-Saharan Africa countries.

Although both Nigeria and Chad having high burden of maternal mortality, the former country had relatively higher SBA as compared with Chad. The finding is consistent with studies conducted elsewhere [36–38]. This difference may be attributed to Nigeria's relatively stronger health infrastructure, broader health workforce availability, and more extensive policy and programmatic efforts aimed at improving maternal health services. For example, Nigeria has implemented national strategies such as the Midwives Service Scheme and the Maternal, Newborn, and Child Health (MNCH) Programme, which aim to increase SBA in underserved areas [39]. In contrast, the lower prevalence of skilled birth attendance in Chad might be because of shortage of trained health professionals and highly reliance on traditional birth attendance; studies revealed that only 24.3% of pregnancy and childbirth had been attended by skilled birth professionals [40,41]. Multiple indicator cluster surveys also implied that 78% of the women were delivered out of the health facility in Chad [42]. Therefore, strong attention by the government and non-governmental organizations is needed in Chad.

By employing multilevel analysis, we uncovered the interplay between individual and community-level factors, offering actionable insights for context-specific maternal health programming. Those reproductive-age women who had a history of ANC visits had higher odds of SBA than those reproductive-age women who had no ANC visit. This finding is similar to studies accompanied in Nepal [43], and Tanzania [44]. This is because women can use multiple information from ANC services about their pregnancy, which in turn enables them to make informed decisions about where to give birth. ANC visit can help the mothers to ascertain their pregnancy status, complications, and support the pregnant women to deliver at the health facility.

Similarly, the odds of skilled birth attendance were higher among study participants who were from high-proportion of community level education than study participants who were from low proportion of community-level education. Consistent with studies conducted in Ethiopia [45], Pakistan [46], and Uganda [47]. The possible justification for the higher SBA among educated women might be that education inspires mothers to be more concerned about their health and have more autonomy, which improves their ability and self-determination to make their own health decisions, ultimately leading to a tendency to more actively seek health care. A woman's ability to afford medical health care is also enhanced by education. Most of the time, educated women are expected to be financially empowered and have better opportunities to attend skilled births [48]. The finding reinforces the critical role of education in shaping maternal health-seeking behavior. Beyond confirming earlier studies, it highlights a persistent policy gap in targeting educational empowerment in maternal health strategies, particularly, for women in rural or impoverished communities. This suggests that multisectoral approaches integrating female education programs with maternal health interventions may yield higher skilled birth coverage.

Findings of this study also demonstrate that the odds of SBA are higher among reproductive-age women who are from middle- and rich-wealth category households as compared with poor-wealth category households. Moreover, lower SBA was observed among low proportions of community-level poverty as compared with high proportions of community-level poverty as reported in sub-Saharan Africa [49] and Ghana [50]. This might be due to their capability to pay for medical and transportation costs; people with good economic status may be more likely to pursue healthcare and make healthcare decisions self-sufficiently [51]. In spite of dispensing skilled birth services at no cost in low-income countries, indirect costs, such as transport and losing wages, can sometimes be higher than direct costs, resulting in underutilization of SBA among poor individuals [52].

The finding of this research revealed that the odds of SBA were observed to be higher among reproductive-age women who had educated primary and higher education as compared with non-educated reproductive-age women. These findings are consistent with the other findings [48] who reported that maternal education significantly improved access to skilled birth services in sub-Saharan Africa, emphasizing that educated women are more likely to be informed about maternal health services and empowered to seek facility-based care.

The odds of SBA were higher among media-exposed participants as compared with their counterparts. Similar to studies conducted in Cameroon [52]. The possibility might be that having media exposure may increase access to information related to maternal SBA. It is also potential that mass media are; in effect, circulating information, which might be facilitating behavioral modifications that allow mothers to accept and use maternal health services [53]. Hence, mass media-produced information is readily understandable by all mothers, which allows promotion and education on maternal health issues to be relatively simple [54].

Lower odds of SBA were observed among reproductive-age women who were from rural places of residence as compared with urban resident reproductive-age women. This is similar with studies conducted in Cameroon [55]. Higher SBA was also observed among reproductive-age women who had not a big problem with distance to the health facility as compared with their counterparts. The possible reason might be the far distance of the health facilities in rural areas [55–57]. Hence, expanding healthcare services in rural areas and improving access to health facilities for rural populations are essential. The finding underscores broader challenges related to health system equity and spatial access areas that require targeted infrastructure investment in both countries, particularly in Chad.

**Strengths and limitations**

The study has the following strengths and limitations. As strengths, the study has been conducted using a large, nationally representative sample size with standardized, representative and validated tools to collect data. Therefore, this finding is generalizable to countries with extremely high maternal mortality, specifically, in Chad and Nigeria. As a result of the study's multilevel approach, the clustering effect of DHS data has been considered, and basic indicators of institutional delivery have been assessed at the individual and community levels. Regarding limitations, the cross-sectional nature of DHS data limits causal inference, and that unobserved variable (e.g., cultural norms, transportation availability) may bias the associations observed. In addition to recall bias, social desirability bias may also be present since respondents were asked about five-year events prior to the survey. There may be other indicators considered as factors for SBA. Therefore, we recommended the future researchers to conduct SBA on women aged 15–49 with a mixed-methods approach.

**Conclusion**

Only 41.98% of reproductive-age women had SBA in Chad and Nigeria. Being educated, wealthy household category, media exposure, urban residence, and distance from the health facility were significantly associated with SBA. To help improve maternal health, governments should make sure more women can access education, get better antenatal care, and afford delivery services, especially, by expanding health insurance. Setting up mobile outreach teams for remote communities and running local awareness campaigns can also make a big difference by helping families understand the importance of giving birth in health facilities. These steps are backed by our findings and are key to meeting Sustainable Development Goal 3.1, which aims to reduce global maternal deaths to fewer than 70 per 100,000 live births by 2030. Connecting these efforts to national strategies will help move the needle and bring safer childbirth to more women.

**Supporting information**

**S1 Data. Data used for analysis.**
(CSV)

**Author contributions**

**Conceptualization:** Wubshet Debebe Negash, Henok Dessie Wubneh.

**Data curation:** Wubshet Debebe Negash, Henok Dessie Wubneh.

**Formal analysis:** Wubshet Debebe Negash, Henok Dessie Wubneh.

**Software:** Wubshet Debebe Negash.

**Validation:** Wubshet Debebe Negash.

**Visualization:** Wubshet Debebe Negash.

**Writing – original draft:** Wubshet Debebe Negash.

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
