## [Decision Letter · Decision Letter 0]

26 Aug 2025

PGPH-D-25-02012

Multilevel analysis of skilled birth attendance and its associated factors in extremely high maternal mortality countries

Dear Dr. Wubshet Debebe Negash,

Thank you for submitting your manuscript to PLOS Global Public Health. After careful consideration, we feel that it has merit but does not fully meet PLOS Global Public Health’s publication criteria as it currently stands. Therefore, we invite you to submit a revised version of the manuscript that addresses the points raised during the review process.

We look forward to receiving your revised manuscript.

Kind regards,

Jayanta Kumar Bora, PhD

Academic Editor

Journal Requirements:

Additional Editor Comments (if provided):

Reviewers' comments:

Reviewer's Responses to Questions

**Comments to the Author**

1. Does this manuscript meet PLOS Global Public Health’s publication criteria?

Reviewer #1: Yes

Reviewer #2: Partly

2. Has the statistical analysis been performed appropriately and rigorously?

Reviewer #1: Yes

Reviewer #2: Yes

3. Have the authors made all data underlying the findings in their manuscript fully available (please refer to the Data Availability Statement at the start of the manuscript PDF file)?

Reviewer #1: No

Reviewer #2: Yes

4. Is the manuscript presented in an intelligible fashion and written in standard English?

Reviewer #1: Yes

Reviewer #2: Yes

Reviewer #1: Manuscript Number: PGPH-D-25-02012 Review Report_SBA

Title:-Multilevel analysis of skilled birth attendance and its associated factors in extremely high maternal mortality countries

Strength

• It shows the urgency of global and national level intervention to improve SBA and MMR in extremely high maternal mortality countries

• It indicate the statistical methods used to analyze DHS data due to it hierarchical nature

Weakness

• The word “extremely high maternal mortality countries” make the title vague in terms of geographic coverage; better if the two counties are mentioned (Chad and Nigeria)

• the orders in the title needs improvement; for decision or policy makers it is better if the outcome comes first rather than statistical methods “Outcome → Determinants → Context → Method

• Multilevel analysis of what type of data? cross sectional study or DHS data better if it indicated in the title “Multilevel analysis of DHS data”

Overall Recommendation and title suggestion

Better if the authors rephrase it again “Skilled birth attendance and its associated factors in Chad and Nigeria: A multilevel analysis of DHS data”

Trends in maternal mortality estimates 2000 to 2023 report; Estimates by WHO, UNICEF, UNFPA, World Bank Group and UNDESA / Population Division

The report said that in 2023, no countries were estimated to have had “extremely high” maternal mortality; the data for the Nigeria and Chad are 993 and Chad 748 MMR respectively. Your reference number 17 about the UNICEF data about maternal mortality also showed this number

Abstract

Strength

• Objectives, Methods, Results, and Conclusions clearly included

• Objective: it is well stated

• Methods: use of DHS data and statistical methods well mentioned (CI and P-value)

• Results: Key findings are well presented with adjusted odds ratios and confidence intervals.

• Conclusion: actionable recommendations are included for respective bodies

Weakness

• Lines 16-17:- the data in the introduction lacks a specific reference year; please indicate the year or time frame for this global estimate. The data of the two countries or actual MMR are less than 1000 in 2023 report; better if precise figures are used to avoid potential misinterpretation. Globally, an estimated 260 000 women died from a maternal cause in 2023

• “The sentence from line 19-21 is redundant. It is already addressed in the aim.

• The phrase “extremely high maternal mortality countries” is repeated three times in the background paragraph. Better if removed or rephrased (line 18, 20 & 23)

Line 24:- It does not clearly shows the survey year of the two countries. Please specify the years of DHS data

The word “expected” is not strong for conclusion or recommendation for the government. It should be mandatory priority for national health systems

Background

Line 49:- Need reference year as indicated above. Providing the reference year for each data point will improve transparency.

Line 53-55:- the sentence globally, in 1990, the maternal mortality rate was 385 deaths per 100,000 live births, but in 2015, it was 216 deaths per 100,000 live births. I think this data is outdated and better to use update data from the latest WHO report data to improve the timeliness and accuracy of the introduction. (The global MMR in 2023 was estimated at 197 maternal deaths per 100 000 live births)

Line 73-75: The sentence “However, this goes in the opposite direction in Chad and Nigeria, where more than 1000 MMR per 100,000 live births still continued (17)”. I think the data of the two countries cited in reference 17 appears to be inconsistent. The data for the year 2023 indicate Nigeria 993 and Chad 748. The data from 2017 to 2020 which is Chad 1063 and Nigeria 1047 cited in reference (10, 11). If it for consistency with DHS survey year clarifying this will improve data transparency and ensure consistency between cited statistics and referenced sources

The term “skilled birth attendance” and its abbreviation “SBA” are used interchangeably across the manuscript. This inconsistency may confuse readers and disrupt the flow of the text better to make the it uniform

Methods

Line 134: Please verify the source and specify the reference year for the >1000 figure

Line 134: The importance of Kids Record (KR) dataset to extract factors associated with SBA was mentioned. It is better if they mention its importance beyond identifying factors.

Line 137: Please specify the years of DHS survey used for Chad and Nigeria in this study.

Study Variable

Media Exposure

Line 150-151: Instead of repeating the above exposure sentence you can rephrase it as “those with no access to any were considered unexposed.

Modeling approaches

Line 179: Better if you explain the importance of each model in the analysis of this study. Why we use the null model? Example the null model includes only random intercepts and used to assess baseline cluster level variance in SBA

Line 179: What is the role of CCI in this study beyond indicating the presence of random effects?

Result

Line 179: Table 1: *Media exposure what (*) these indicate? It is not explained under the table

Line 182: Better if the prevalence of individual counties mentioned for ease of understanding in the sentence rather than presenting pooled estimates

Line 195: I think it is better to include “AOR” in this section since you are reporting for the first time in this part

Line 195-208: The phrase “reproductive-age women” is repeated eight times within a single paragraph—which disrupts the flow and readability of the text. I recommend if they use pronouns or simplified phrase

Discussion

Line 210: During the discussion of the finding based on DHS survey conducted at different time (Chad 2014–15, Nigeria 2018 if the years are correct) the observed differences in skilled birth attendance between Chad and Nigeria should be interpreted with caution due to the temporal gap in DHS survey implementation. Time variation between surveys can introduce confounding effects that influence health system performance indicators, including skilled birth attendance

Line 210: the phrase Sub Saharan Africa and the abbreviation SSA should be used consistently

Line 239: In most of academic writings “odds” is treated as a plural noun. I think better to consider the grammar across the manuscript. Example Odds of skilled birth attendance were..."

Line 239: Why different citation style is used in this sentence? Consistency in citation is mandatory throughout the entire document. In academic writing, using different citation styles within the same manuscript is generally not acceptable

Line 239: It would be more effective to begin the discussion with the factor showing the highest odds—such as antenatal care (ANC) visits—which is also the most influential predictor of skilled birth attendance (SBA) . It helps to develop priority intervention to improve SBA

Line 240: The discussion of odds ratio started with educational factors; from the lower it is at third stage. I think it is better if you start with the most influential factors first (e.g., starting with ANC visit: OR 5.56) this may help the policy makers to give priority for intervention based on the study findings

Odds of Skilled Birth Attendance (Increasing Order)

Factor Odds Ratio (OR) 95% Confidence Interval (CI)

Middle wealth status 1.37 (1.23 – 1.52)

Media exposure 1.5 (1.38 – 1.63)

Primary education 1.77 (1.62 – 1.95)

Rich wealth status 2.11 (1.87 – 2.38)

High community education level 2.73 (2.26 – 3.29)

Secondary education 4.06 (3.59 – 4.57)

ANC attendance 5.56 (5.03 – 6.14)

Line 333: Data Availability Statement: The data used in this study are publicly available from the Demographic and Health Surveys (DHS) Program through a formal request process through the DHS website: https://dhsprogram.com/data. I think this should be clearly stated.

My question; is the sentence appropriate for DHS survey data from ethical point of view?

Example of the Trend of MMR in 2023

For the sentence “International Children’s Fund (UNICEF) reported that South Sudan, Chad, and Nigeria had extremely high maternal mortality rates (>1000) in 2023. The picture may be used as reference

African countries with the highest maternal mortality rate in 2023(deaths per 100,000 live births)

Overall, the manuscript is well written; however, if the above points are incorporated, it will further enhance its clarity

Reviewer #2: Major comments:

• Why is the prevalence of skilled birth attendance almost 30% higher than previously reported?

• There is a contradiction in that although the prevalence of SBA is higher in Nigeria (Fig. 1), table 3, model 3, suggests that the adjusted odds for SBA is higher in Chad than Nigeria. This needs to be discussed.

• There appears to be no discussion of the effects of community level variables. Then what’s the point of including these in your models? Would any of the conclusions have changed if you had used only individual level variables in your model?

• Is SBA self reported by mothers? If yes, can there be variations in respondents understanding of what SBA means? If yes, how does this affect results?

Technical comments:

• How was the Kids Record linked to the survey data? Was there a unique identifier?

• How were the community level clusters created? Provide details on geographic location and clustering algorithm.

• Howe correlated are the predictors? Is multicollinearity a problem, i.e. are adjusted odds sensitive to the presence of correlated variables in the model?

• What do the weights in modelling depend on? Provide more details.

• What is the bivariable analysis? Having model equations would be helpful for understanding.

• Consider the use of AIC instead of deviance, as models have different numbers of parameters.

Minor comments:

• Keep table headers on same page as table contents.

• Table 3: where is the bivariable model?

• Is Fig. 1 necessary? Can’t that data be incorporated in Table 2?

**Do you want your identity to be public for this peer review?** For information about this choice, including consent withdrawal, please see our Privacy Policy

Reviewer #1: No

Reviewer #2: No

---

## [Decision Letter · Decision Letter 1]

11 Nov 2025

PGPH-D-25-02012R1

Skilled birth attendance and its associated factors in Chad and Nigeria: A multilevel analysis of DHS data

Dear Dr. Wubshet Debebe Negash,

Thank you for submitting your manuscript to PLOS Global Public Health. After careful consideration, we feel that it has merit but does not fully meet PLOS Global Public Health’s publication criteria as it currently stands. Please review the reviewer comments carefully and address it.Therefore, we invite you to submit a revised version of the manuscript that addresses the points raised during the review process.

We look forward to receiving your revised manuscript.

Kind regards,

Jayanta Kumar Bora,PhD

Academic Editor

Journal Requirements:

Additional Editor Comments (if provided):

Reviewers' comments:

Reviewer's Responses to Questions

**Comments to the Author**

Reviewer #1: All comments have been addressed

publication criteria?

Reviewer #1: Yes

3. Has the statistical analysis been performed appropriately and rigorously?

Reviewer #1: Yes

4. Have the authors made all data underlying the findings in their manuscript fully available (please refer to the Data Availability Statement at the start of the manuscript PDF file)?

Reviewer #1: Yes

5. Is the manuscript presented in an intelligible fashion and written in standard English?

Reviewer #1: Yes

Reviewer #1: Reviewed Summary– Manuscript PGPH-D-25-02012R1

I have reviewed the revised manuscript titled “Skilled birth attendance and its associated factors in Chad and Nigeria: A multilevel analysis of DHS data.” The authors have submitted a detailed point-by-point response and made substantial revisions to the manuscript. Below is a summary of my evaluation based on the comments I previously raised:

The revised manuscript demonstrates strong improvements in clarity, methodological rigor, and contextual relevance. The title and abstract are well-structured; country selection and survey years are justified. Definitions, variable coding, and data access are clearly explained. The multilevel modeling approach is sound, with appropriate weighting, model comparison, and interpretation of random effects. Tables and formatting are improved. Key predictors of SBA ANC visits, education, wealth, media exposure, and residence are well contextualized. Ethical procedures are documented, and policy recommendations are relevant and actionable.

However, the following points require further revision. Please consider reducing redundancy by using synonyms or restructuring sentences to enhance clarity and maintain a professional academic style.

Comment 1: The phrase "skilled birth attendance" appears excessively (7 times in the abstract, 62 times overall), which may affect readability. Please try to address all redundant words like skilled birth attendance, extremely high maternal mortality

Comment 2: The phrase “extremely high maternal mortality” is overused (20 times); consider varying it for clarity. Also, “SBA” is underused despite being defined. After defining the acronym, consider using "SBA" more consistently to simplify the text and avoid repetition of the full phrase. Please try to address all redundant words

Comment 3: The term “reproductive-age women” appears 3 times in the abstract and 23 times in the full manuscript. Consider reducing repetition by using alternatives such as “women of childbearing age,” “target population,” or “study participants” where appropriate to enhance readability and stylistic variation

Overall Assessment:

The authors have satisfactorily addressed all the previous comments. The manuscript is now substantially improved in terms of clarity, methodological rigor, and contextual relevance. I recommend proceeding with editorial consideration.

**Do you want your identity to be public for this peer review?** For information about this choice, including consent withdrawal, please see our Privacy Policy

Reviewer #1: No

---

## [Editor Report · Decision Letter 2]

26 Nov 2025

Skilled birth attendance and its associated factors in Chad and Nigeria: A multilevel analysis of DHS data

PGPH-D-25-02012R2

Dear Wubshet Debebe Negash,

We are pleased to inform you that your manuscript 'Skilled birth attendance and its associated factors in Chad and Nigeria: A multilevel analysis of DHS data' has been provisionally accepted for publication in PLOS Global Public Health.

Best regards,

Jayanta Kumar Bora,PhD

Academic Editor